# Characterization of Lysine Acetyltransferase Activity of Recombinant Human ARD1/NAA10

**DOI:** 10.3390/molecules25030588

**Published:** 2020-01-29

**Authors:** Tam Thuy Lu Vo, Ji-Hyeon Park, Eun Ji Lee, Yen Thi Kim Nguyen, Byung Woo Han, Hien Thi Thu Nguyen, Kyo Cheol Mun, Eunyoung Ha, Taeg Kyu Kwon, Kyu-Won Kim, Chul-Ho Jeong, Ji Hae Seo

**Affiliations:** 1Department of Biochemistry, Keimyung University School of Medicine, Daegu 42601, Korea; volutam@gmail.com (T.T.L.V.); nguyenthithuhien.th@gmail.com (H.T.T.N.); mun@dsmc.or.kr (K.C.M.); hanne.md@gmail.com (E.H.); 2Departments of Radiology and Neurology, Massachusetts General Hospital and Harvard Medical School, Charlestown, MA 02129, USA; jihyeon000@gmail.com (J.-H.P.); gulpgulp89@gmail.com (E.J.L.); 3College of Pharmacy and Research Institute of Pharmaceutical Sciences, Seoul National University, Seoul 08826, Korea; kimyen@snu.ac.kr (Y.T.K.N.); bwhan@snu.ac.kr (B.W.H.); qwonkim@snu.ac.kr (K.-W.K.); 4Department of Immunology, Keimyung University School of Medicine, Daegu 42601, Korea; kwontk@dsmc.or.kr; 5College of Pharmacy, Keimyung University, Daegu 42601, Korea

**Keywords:** Arrest defective 1 (ARD1), acetylation, N(alpha)-acetyltransferase 10 (NAA10), lysine acetyltransferase (KAT), N-terminal acetyltransferase (NAT)

## Abstract

Arrest defective 1 (ARD1), also known as N(alpha)-acetyltransferase 10 (NAA10) was originally identified as an N-terminal acetyltransferase (NAT) that catalyzes the acetylation of N-termini of newly synthesized peptides. After that, mammalian ARD1/NAA10 expanded its’ role to lysine acetyltransferase (KAT) that post-translationally acetylates internal lysine residues of proteins. ARD1/NAA10 is the only enzyme with both NAT and KAT activities. However, recent studies on the role of human ARD1/NAA10 (hARD1/NAA10) in lysine acetylation are contradictory, as crystal structure and in vitro acetylation assay results revealed the lack of KAT activity. Thus, the role of hARD1/NAA10 in lysine acetylation is still debating. Here, we found a clue that possibly explains these complicated and controversial results on KAT activity of hARD1/NAA10. Recombinant hARD1/NAA10 exhibited KAT activity, which disappeared soon in vitro. Size-exclusion analysis revealed that most recombinant hARD1/NAA10 formed oligomers over time, resulting in the loss of KAT activity. While oligomeric recombinant hARD1/NAA10 lost its ability for lysine acetylation, its monomeric form clearly exhibited lysine acetylation activity in vitro. We also characterized the KAT activity of hARD1/NAA10 that was influenced by several experimental conditions, including concentration of reactants and reaction time. Taken together, our study proves that recombinant hARD1/NAA10 exhibits KAT activity in vitro but only under accurate conditions, including reactant concentrations and reaction duration.

## 1. Introduction

Post-translational modifications (PTMs) refer to the chemical alterations in proteins following syntheses to regulate their activities and cellular functions. Acetylation is one of the most important PTMs characterized with the transfer of an acetyl group from acetyl-coenzyme A to either the N-terminal amino acid or a lysine residue of protein, and is catalyzed with N-terminal acetyltransferases (NATs) and lysine acetyltransferases (KATs), respectively.

N-terminal acetylation catalyzed by NATs is one of the most common protein modifications in eukaryotes, affecting about 80% human proteins and 60% yeast proteins [1,2]. In general, NATs acetylate N-terminal residues of newly synthesized proteins from ribosomes in an irreversible manner. N-terminal acetylation is known to be closely related to protein stability, interaction, and localization.

In contrast to N-terminal acetylation, lysine acetylation catalyzed by KATs is reversibly regulated by lysine deacetyltransferases (KDACs) that remove acetyl groups from lysine residues in proteins [3]. While acetylation neutralizes the positive charge on lysine residues, deacetylation recovers it, thereby causing a change in electronic and conformational properties of proteins. As a result, these changes impact the interaction, enzymatic activity, and biological functions of proteins. Acetylation and deacetylation of lysine residues serve as the switches that turn-on and turn-off the cellular signal pathways and regulate diverse biological events. Any unbalance between lysine acetylation and deacetylation results in the improper regulation of biological processes and may cause various types of human diseases such as cancer and neurodegeneration.

Arrest defective 1 (ARD1) is the only enzyme known so far to exhibit both NAT and KAT activities. ARD1 was originally identified as NAT in yeast. In 1985, Whiteway and Szostak found that the mutation in ARD1 could cause defects in the modulation of cell cycle in *Saccharomyces cerevisiae* [4]. Thereafter, ARD1 was found to be highly conserved in eukaryotes and involved in a wide range of biological processes [5]. Surprisingly, mammalian ARD1, also known as NAA10, has been found to exhibit KAT as well as NAT activities. Aside from catalyzing N-terminal protein acetylation reactions, mammalian ARD1/NAA10 has been reported to acetylate internal lysine residues of various proteins, including Hypoxia-inducible factor 1-alpha (HIF-1α) [6], β-catenin [7], Runt-related transcription factor 2 (RUNX2) [8], methionine sulfoxide reductase A (MSRA) [9], myosin light chain kinase (MLCK) [10], androgen receptor [11], heat shock protein 70 (Hsp70) [12], and phosphoglycerate kinase 1 (PGK1) [13]. During the lysine acetylation of these substrate proteins, hARD1 regulates a wide range of cellular functions, including cell cycle, apoptosis, migration, stress response, and differentiation [14].

However, some studies have presented conflicting results about the lysine acetylation activity and claimed the absence of KAT activity for human ARD1/NAA10 (hARD1/NAA10). Recombinant hARD1/NAA10 (rhARD1/NAA10) protein was unable to acetylate HIF-1α but catalyzed its own N-terminal residue in vitro, suggestive of its NAT activity but not KAT activity [15]. Similar to this study, Magin et al. showed that rhARD1 failed to acetylate the lysine residues of MSRA, MLCK, and RUNX2, suggesting that these proteins were chemically acetylated in vitro, but not by rhARD1 [16]. Crystallographic structure analysis also showed that hARD1/NAA10 lacks enough space to perform lysine acetylation of substrate proteins, implicative of its role only as NAT rather than KAT [17,18]. Based on these controversial results on the KAT activity of hARD1/NAA10, some studies suggest that the KAT activity of hARD1 could be activated under some specific conditions or was very weak in vitro [19,20]. Therefore, further studies are warranted to reconcile these opposing views on the KAT activity of hARD1/NAA10. Here, we found a clue for the explanation of the controversial data on hARD1/NAA10 KAT activity *in vitro*. We characterized the KAT activity of rhARD1/NAA10 in vitro and found that it easily disappeared during protein purification. Most rhARD1/NAA10 aggregated over time during purification, thereby losing its KAT activity. We also observed that rhARD1/NAA10 K136R mutant, which lacks KAT activity, exposed strong NAT activity, suggesting that NAT activity and KAT activity are possibly regulated independently. Furthermore, we clarified the specific experimental conditions that are suitable for investigating KAT activity of hARD1/NAA10 in vitro. Under these conditions, hARD1/NAA10 was found to retain its KAT activity. Our results show that the lysine acetylation of substrate proteins is not an artificial chemical reaction but is directly mediated via rhARD1/NAA10 KAT activity.

## 2. Results

### 2.1. KAT Activity of rhARD1/NAA10 Is Lost During Dialysis

In a previous report, we described the autoacetylation of rhARD1/NAA10 in vitro [21]. However, autoacetylation activity of purified hARD1 failed to last long; thus, only freshly purified hARD1/NAA10 was subjected to in vitro acetylation assay [21]. Based on this experience, we hypothesized that the lysine acetylation of rhARD1/NAA10 may be unstable and easily disappear during purification. We evaluated the time for which the lysine acetylation activity of rhARD1/NAA10 could last by collecting rhARD1/NAA10 from different steps of purification using agarose beads (Figure 1A). First, we collected rhARD1/NAA10 bound to the beads, and then collected it again immediately after elution. Finally, we collected hARD1/NAA10 after dialysis, the final step of protein purification. We have previously reported that hARD1/NAA10 undergoes autoacetylation, which was critical to stimulate its KAT activity [21]. To investigate the KAT activity of rhARD1/NAA10 collected during different steps, we evaluated their autoacetylation states. As a result, the lysine acetylation activity of hARD1/NAA10 was found to be well maintained until the elution step but dramatically diminished after overnight dialysis (Figure 1B). To evaluate the hARD1/NAA10-mediated lysine acetylation of substrate protein, we performed in vitro acetylation using Hsp70, a known substrate protein of hARD1/NAA10 [12]. Consistent with the results of autoacetylation state, rhARD1/NAA10 collected at the final step of purification was unable to acetylate Hsp70 (Figure 1C). These results indicate that rhARD1 has the tendency to lose its lysine acetylation activity in vitro.

### 2.2. rhARD1/NAA10 Is Aggregated During Purification

To investigate the reason underlying the instability of the lysine acetylation activity of ARD1 in vitro, we purified and analyzed the oligomeric state of rhARD1/NAA10 with size-exclusion chromatography (Figure 2A). We evaluated the change in the oligomeric state of hARD1/NAA10 in response to the purification process by collecting it at different steps and comparing the oligomeric states. rhARD1/NAA10 was collected after its elution from Ni-NTA affinity column and again after the anion-exchange step, the final step of purification. The size-exclusion chromatography profile of the eluted protein from Ni-NTA column revealed its various oligomeric states, including monomeric, dimeric, and high oligomeric forms (Figure 2B). After anion-exchange purification, size-exclusion chromatography showed that rhARD1/NAA10 mainly existed in a high oligomeric state and had only a few monomers (Figure 2B). These results indicate that the proportion of the monomeric form of rhARD1/NAA10 after affinity purification was higher than that after anion-exchange purification, possibly explaining the neglected acetyltransferase activity of rhARD1/NAA10 in vitro after dialysis.

To determine the KAT activity of hARD1/NAA10 in different states, the monomeric and highly oligomeric forms of hARD1/NAA10 were subjected to in vitro acetylation in the presence of the substrate Hsp70. As a result, the monomeric form mostly exhibited a strong ability to acetylate itself, whereas the highly oligomeric form lost its autoacetylation activity. As a consequence, Hsp70 could only be acetylated by the monomeric rhARD1/NAA10 form but not by the oligomeric form (Figure 2C). Thus, rhARD1/NAA10 loses its KAT activity during the purification process, owing to the formation of high oligomers. To identify whether hARD1/NAA10 in different states exposes the similar pattern of KAT and NAT activities, various oligomeric forms of rhARD1/NAA10 were subjected to DTNB-based assay for quantifying the NAT activity of hARD1/NAA10 with presence of EEEIA or DDDIA peptides representing N-termini of γ-actin and β-actin respectively. The results showed that even though oligomeric form of hARD1/NAA10 lost its autoacetylation activity and KAT activity towards its substrate Hsp70, but still exhibited the NAT activity towards the EEEIA and DDDIA peptide substrates (Figure 2D).

### 2.3. rhARD1/NAA10 Mediates Site-Specific Lysine Acetylation of Substrate That Is Distinctive from the Non-Specific Chemical Reaction

Although hARD1/NAA10 was reported to have both NAT and KAT activities, whether it could catalyze protein lysine acetylation is still debatable. Researchers have argued about the KAT activity of hARD1/NAA10 because rhARD1 exhibits NAT activity but not KAT activity in vitro [15,16]. In particular, a recent study suggested that rhARD1/NAA10-mediated in vitro acetylation may be an artificial reaction and that substrates may be acetylated chemically, but not enzymatically. Considering these conflicting results, we characterized the enzymatic activity of rhARD1/NAA10 in vitro. To differentiate between chemical and enzymatic reactions, we performed an in vitro acetylation assay using wild-type (WT) and mutant rhARD1/NAA10. We used two mutant forms, K136R and dominant-negative (DN) mutant, that lack acetyltransferase activity [21]. hARD1/NAA10 is known to acetylate its K136 lysine residue, and K136 acetylation is important to stimulate its KAT activity [21]. DN mutant includes two mutations R82A and Y122F, which inhibit the binding of acetyl-CoA to hARD1/NAA10 and consequently suppresses its acetyltransferase activity [21]. We performed in vitro acetylation assay with these two mutants, and found that only the wild-type rhARD1/NAA10 could perform autoacetylation, while K136R and DN mutants failed to acetylate themselves (Figure 3A). In addition, lysine acetylation of Hsp70 was also stimulated only with the wild-type rhARD1/NAA10 but not with K136R mutant nor DN rhARD1/NAA10 (Figure 3B). Similarly, only rhARD1/NAA10 WT but neither ARD1/NAA10 K136R mutant nor DN exhibited the catalytic activity on β-catenin (Figure 3C). These results indicate that the in vitro lysine acetylation mediated by hARD1/NAA10 was directly controlled by its enzymatic activity rather than any chemical reaction. Furthermore, the hARD1/NAA10-mediated catalysis of Hsp70 was abolished with K77R mutation in Hsp70, wherein K77 is known as a target site of lysine acetylation [12] (Figure 3D). These data indicate that hARD1/NAA10-mediated lysine acetylation is a site-specific reaction rather than a non-specific artificial reaction. Then we checked whether the NAT activity of hARD1/NAA10 could be affected by the regulation of autoacetylation of hARD1/NAA10. The rhARD1/NAA10 WT and non-acetylated K136R mutant and acetyltransferase dead DN mutant were applied to the Nt-acetyltransferase assay in the presence of EEEIA and DDDIA peptides. Interestingly, while the acetyltransferase dead DN mutant of hARD1/NAA10 almost lost its NAT activity, the non-acetylated K136R mutant exposed N-terminal acetyltransferase capacity as strongly as the hARD1/NAA10 WT did (Figure 3E).

### 2.4. Lysine Acetylation Activity of hARD1/NAA10 Is Dependent on Concentration of Reactants

We investigated how the KAT activity of hARD1/NAA10 was affected by in vitro reaction conditions such as dose of reaction components. We first examined the effect of hARD1/NAA10 concentration on lysine acetylation of substrate. As shown in Figure 4A, Hsp70 acetylation increased with an increase in hARD1/NAA10 concentration, confirming that Hsp70 acetylation is regulated by activity of hARD1/NAA10 that is distinct from chemical reaction. As a previous report showed that high concentration of acetyl-CoA chemically induced lysine acetylation in vitro [22], we examined the effect of acetyl-CoA concentration on hARD1/NAA10-mediated lysine acetylation in vitro. As shown in Figure 4B, Hsp70 acetylation increased in the presence of acetyl-CoA in a concentration-dependent manner. However, we failed to observe any Hsp70 acetylation in the absence of hARD1/NAA10. These data indicate that Hsp70 acetylation is directly mediated by hARD1/NAA10 but not by high concentrations of acetyl-CoA. High concentration of acetyl-CoA only boosts the lysine acetylation activity of hARD1/NAA10 but may not stimulate chemical acetylation. To confirm that acetylation is a hARD1/NAA10-mediated enzymatic reaction, but not an artificial chemical reaction, we destroyed hARD1/NAA10 structure by heating it at 95 °C and then performed in vitro acetylation assay. As shown in Figure 4C,D, heat-inactivated hARD1/NAA10 could not mediate any lysine acetylation reaction as compared with the unheated hARD1/NAA10, suggesting that the accurate structure of hARD1/NAA10 is indispensable to stimulate its lysine acetylation activity in vitro. Together these results not only highlight the KAT activity of rhARD1/NAA10 but also show that rhARD1/NAA10-mediated in vitro acetylation is not an artificial chemical reaction.

### 2.5. Limited Reaction Conditions Are Needed to Distinguish the hARD1/NAA10-Mediated Lysine Acetylation Reaction from Non-Specific Chemical Reaction

We examined the effect of reaction time on in vitro acetylation using rhARD1/NAA10, as many studies have performed reactions for different durations. We compared the hARD1/NAA10-mediated Hsp70 acetylation at various reaction time points. Until 2 h, no Hsp70 acetylation was observed without rhARD1/NAA10, while rhARD1/NAA10-mediated Hsp70 acetylation increased in a time-dependent manner (Figure 5A). However, Hsp70 acetylation was observed even in the absence of rhARD1/NAA10 after 3 h (Figure 5B). These results suggest that reactions over 2 h could induce chemical acetylation; thus, this condition is unsuitable for the measurement of the lysine acetylation activity of rhARD1/NAA10. To confirm that the limited reaction time was critical to distinguish the real enzymatic lysine acetylation reaction from the artificial chemical acetylation reaction, we analyzed autoacetylation of hARD1/NAA10 using WT and DN hARD1/NAA10 recombinants at different reaction times (Figure 5C). Consistent with the result of Hsp70 acetylation, we observed different acetylation levels between WT and DN hARD1 recombinants at 1 h, and this difference was not clear at 4 h, confirming the importance of limited reaction time for hARD1/NAA10-mediated lysine acetylation in vitro (Figure 5C). It is reported that ε-amino group should be deprotonated prior to a nucleophilic attack in acetylation reaction, suggesting alkaline is a favorable condition for acetylation reaction [22]. Moreover, though the intracellular cytosolic and nuclear pH of a cell is in the range 7.0–7.3 [23], the pH varies in different organelles of a cell [24]. To check to effect of pH on in vitro acetylation assay, we subjected the rhARD1/NAA10 and Hsp70 to the various pH of the reaction. The acetylation of Hsp70 was not detected at pH below 7.5, while there was no difference in the acetylation of Hsp70 in the presence or absence of hARD1/NAA10 at pH 8.5 and 9.0 (Figure 5D). However, the difference in the acetylation of Hsp70 with or without rhARD1/NAA10 at pH 8.0 (Figure 5D,E) suggests the optimal condition for distinguish the enzymatic acetylation and non-specific reaction. In addition, reactions with various enzyme/substrate ratios were also evaluated. The data showed that the difference in the acetylation of Hsp70 with or without rhARD1 could be observed at low ratio of enzyme: substrate up to 1:25 but not at that of higher ratio over 1:25 (Figure 5F).

## 3. Discussion

While hARD1/NAA10 was first identified as a NAT, subsequent studies revealed its KAT activity. Many protein substrates that are acetylated at internal lysine residues by hARD1/NAA10 have been discovered over past decades. Nevertheless, a few studies have doubted the KAT activity of hARD1/NAA10.

Jeong et al. first reported the KAT activity of mammalian ARD1/NAA10 by demonstrating the acetylation of K532 residue of HIF-1α by mouse ARD1/NAA10, which plays an important role in the regulation of tumor angiogenesis [6]. In contrast to this study, subsequent studies revealed the inability of hARD1/NAA10 to acetylate HIF-1α [15]. hARD1/NAA10 was also reported to acetylate K49 residue of MSRA [9] and K225 residue of RUNX2 [8], and consequently, contributes to the regulation of redox stress response and bone development, respectively. However, Magin et al. strongly contradicted these results and suggested the possibility of in vitro chemical acetylation of MSRA and Runx2, instead of the enzymatic activity of rhARD1/NAA10 [16].

We have previously reported the unstable KAT activity of rhARD1/NAA10 in vitro. rhARD1/NAA10 rapidly lost its KAT activity over time [21]. Thus, the KAT activity of rhARD1/NAA10 was detected only with the freshly purified protein in in vitro acetylation assay. In addition, we observed precipitation of rhARD1/NAA10 after dialysis during large-scale rhARD1/NAA10 purification (data not shown). Based on this experience, we assumed that the KAT activity of rhARD1/NAA10 could be very unstable and hence may have led to conflicting results.

In the present study, we characterized rhARD1/NAA10 with respect to its unstable KAT activity. We found that most rhARD1/NAA10 was prone to aggregation during purification, resulting in the loss of its lysine acetylation activity. Modifications in the purification steps could produce enzymatically active rhARD1/NAA10, which displayed KAT activity in vitro. We assume that the stability of hARD1/NAA10 could rely on its binding to other proteins. ARD1/NAA10 predominantly localizes in cell cytosol as a subunit of the stable complex NatA with Naa15p [25,26]. ARD1/NAA10 was recently found to exist independently from the NatA complex in the cytosol and had the ability to perform post-translational acetylation [27]. In addition, ARD1/NAA10 was also found localized in the nucleus independent of Naa15p [25,28]. As ARD1/NAA10 shows NAT activity in NatA complex, it is likely that ARD1/NAA10 has KAT activity in its monomeric form in cells. Our study provides evidence that the monomeric form of hARD1/NAA10 had KAT activity (Figure 2C). How the monomeric hARD1/NAA10 form maintains its stability and exhibits KAT activity in cells, however, remains unknown. The binding of ARD1/NAA10 to Naa15p induces a conformation change in ARD1/NAA10, thereby increasing its NAT activity [17]. There may be conformation changes in the monomeric ARD1/NAA10 form regulated by its interaction with other proteins, resulting in the activation of its KAT activity toward substrates. Recently, Kang et al. evidenced that ARD1/NAA10 is hydroxylated by Factor Inhibiting HIF (FIH) at tryptophan 38 residue under normoxia condition, leading to the open of the catalytic pocket of ARD1/NAA10 that allows the lysine acetylation of HIF-1α [29]. Whether the interaction between ARD1/NAA10 and FIH influences ARD1/NAA10 stability as well as ability to catalyze other substrates besides HIF-1α is still an open question. Additionally, the NAT activity of oligomeric form of hARD1/NAA10 still remained; especially the dimeric form was able to catalyze the N-termini of substrates as strong as monomeric form despite of their loss of KAT activity (Figure 2C,D). Inhibition of hARD1/NAA10 autoacetylation by K136R mutation induced the drop of KAT activity, but not NAT activity (Figure 3E). These observations suggest that NAT and KAT activity might be independently regulated, relying on the interaction partners. Although the structure of ARD1/NAA10 as a subunit form of NatA has been unveiled in several species, that of the monomeric hARD1/NAA10 form is yet identified owing to its aggregation during purification. Further studies on monomeric hARD1/NAA10 structure would provide an insight into the stability and enzymatic activity of hARD1/NAA10.

A line of evidence demonstrates the possibility of chemical lysine acetylation [16]. Therefore, we attempted to differentiate between chemical and enzymatic acetylation. As a result, we conclude that hARD1/NAA10 has lysine acetylation activity *in vitro*. Mutations in hARD1/NAA10 reduced its enzymatic activity and prevented its ability to acetylate substrates. The heat-induced disruption of hARD1/NAA10 structure also diminished its lysine acetylation activity. In addition, acetylation of substrate occurred at specific residues such as K77 of Hsp70 and K136 of hARD1/NAA10. These results suggest that the rhARD1/NAA10-mediated lysine acetylation is not a non-specific chemical reaction, but rather an enzymatic reaction.

We also determined the conditions suitable for in vitro lysine acetylation using rhARD1/NAA10. We found that an acetylation reaction taking place over 3 h was unsuitable because the substrate was strongly acetylated even without hARD1/NAA10, consistent with the results of previous studies. Furthermore, since the acetylation of substrates at pH higher than 8.5 with or without hARD1/NAA10 was incompatible, the in vitro acetylation reaction at pH higher than 8.5 is inappropriate for studying the KAT activity of hARD1/NAA10 though the acetylation reaction is favorable in alkaline condition. Therefore, for the accurate observation and interpretation of in vitro acetylation assay results with hARD1/NAA10, numerous experimental conditions such as the state of recombinant proteins, concentration of enzymes and substrates, and duration of reactions are needed to be considered. In addition, the comparison the acetylation of the substrate in the presence and absence of the enzyme is necessary to study the KAT activity of hARD1/NAA10 because of the basal level chemical acetylation.

Several studies have demonstrated the contribution of the KAT activity of hARD1/NAA10 to tumor development [7,11,13,30]. The development of selective inhibitors of hARD1/NAA10 KAT activity may be potentially useful for cancer therapy. However, there are several obstacles in the discovery of KAT-selective inhibitors of hARD1/NAA10, particularly the unknown structure of monomeric hARD1/NAA10 and inability to detect lysine acetylation efficiency in vitro. Thus, accurate conditions for in vitro acetylation of hARD1/NAA10 should be noted. Here, we provide the characteristics and details of hARD1/NAA10-mediated lysine acetylation that may be useful for the development of hARD1/NAA10 inhibitors.

## 4. Materials and Methods

### 4.1. Antibodies

Anti-acetylated lysine (Lys-Ac) antibody (#9941) were purchased from Cell Signaling Technology. Anti-his antibody (sc-8036), anti-GST antibody (sc-138) were from Santa Cruz.

### 4.2. Plasmid Construction

The full length of cDNA for human ARD1/NAA10 (NCBI Reference Sequence: NM_003491.4) was amplified by using PCR and cloned into the vector pET28a for induction His-tagged ARD1/NAA10 and the vector pGEX4T3 for GST-tagged ARD1/NAA10 recombinant proteins. The gene encoding Hsp70 (NCBI Reference Sequence: NM_005345.6) was acquired from PCR and cloned into the vector pGEX4T3 for induction of GST-tagged Hsp70 recombinant protein. Point mutated mutants of ARD1/NAA10 and Hsp70 were established by mutagenesis kit (#15071, iNtRON, Gyeonggi-do, Korea) according to the manufacturer’s manual as described [12].

### 4.3. Expression and Purification of His-ARD1

BL21 competent *E. coli* cells were transformed with the constructed plasmids and grown in LB media containing kanamycin at 37 °C to OD600 of 0.6–0.8. The culture was induced with 0.5 mM of isopropyl β-d-1-thiogalactopyranoside (IPTG) and grown at 25 °C overnight. Cells were harvested by centrifugation and lysed by sonication (SONICS, CT, USA) for 15 min at 60% amplitude in the buffer containing 20 mM Tris-HCl (pH 7.5), 500 mM NaCl, 10 mM imidazole and 1 mM phenylmethylsulfonyl fluoride.

For a small scale of purification, the lysate was centrifuged and the supernatant was mixed with equilibrated Ni-NTA resins (Invitrogen, Basel, Switzerland) for 2 h at 4 °C. The resins were washed with the washing buffer containing 20 mM Tris-HCl (pH 7.5), 500 mM NaCl, and 20 mM imidazole. The overexpressed protein was eluted with elution buffer (20 mM Tris-HCl (pH 7.5), 500 mM NaCl, and 1 M imidazole). The eluted protein was then subjected into the dialysis membrane (Spectrum Spectra/Por) in dialysis buffer (PBS) at 4 °C overnight.

For the large scale of purification, the supernatant of lysed mixture of proteins was loaded onto a 5-mL HiTrap Chelating HP column (GE Healthcare) pre-charged with Ni^2+^ and equilibrated with buffer containing 20 mM Tris-HCl (pH 7.5), 500 mM NaCl, and 35 mM imidazole. After washing with the buffer used in equilibration, the retained proteins were eluted by addition of an increasing gradient of buffer containing 20 mM Tris-HCl (pH 7.5), 500 mM NaCl, and 1 M imidazole. The eluted fractions were pooled and divided into two fraction pool. The first fraction pool was dialyzed with the buffer containing 20 mM Tris-HCl (pH 7.2) and 30 mM NaCl and loaded onto a 5 mL HiTrap Q HP column (GE Healthcare). The bound proteins were eluted by addition of an increasing gradient of buffer containing 20 mM Tris-HCl (pH 7.2) and 1 M NaCl. The eluted anion-exchange fractions were pooled and loaded onto a HiLoad 16/600 Superdex 200 pg column with AKTA FPLC (GE Healthcare) equilibrated with PBS buffer. The purities of protein fractions were confirmed by SDS-PAGE and native-PAGE. The second fraction pool was directly injected into the HiLoad 16/600 Superdex 200 pg column with AKTA FPLC (GE Healthcare) and eluted in the same PBS buffer as the first fraction pool.

### 4.4. Oligomeric Status Characterization by Size-Exclusion Chromatography

The oligomeric status of hARD1/NAA10 recombinant was determined by gel-filtration chromatography at 20 °C in a HiLoad 16/600 Superdex 200 pg column with AKTA FPLC (GE Healthcare, Pittsburgh, PA, USA) following equilibrium with PBS buffer. The mobile phase was PBS buffer and the flow rate was 1.0 mL/min. The column was calibrated under identical running conditions with the molecular mass protein standards from Gel Filtration Standard (BIO-RAD, Hercules, CA, USA). The standard protein mixture contained: thyroglobulin, 670 kDa; γ-globulin, 158 kDa; ovalbumin, 43 kDa; myoglobin, 17 kDa; and vitamin B12, 1.35 kDa. The eluted proteins were detected by monitoring the absorbance at 280 nm.

### 4.5. Expression and Purification of GST-ARD1/NAA10

Transformed BL21 competent *E. coli* cells with constructed plasmids were grown in LB media supplemented with ampicillin at 37 °C to OD600 of 0.6–0.8 prior to inducing with 0.5 mM of IPTG at 25 °C overnight. Harvested cells by centrifugation were sonicated in the lysis buffer (250 mM Tris-Cl pH 7.5, 500 mM NaCl, 5 mM EDTA and 1% Triton X-100, and 1 mM phenylmethylsulfonyl fluoride). The cleared lysate was incubated with Glutathione-resins (Thermo Fisher Scientific, Rockford, IL, USA) for 2 h at 4 °C, following by three washes with washing buffer. The bound protein was maintained in PBS for later use.

### 4.6. Expression and Purification of GST-Hsp70

GST-tagged Hsp70 construct was expressed in the BL21 competent *E. coli* cells. Cells were grown in LB supplemented with ampicillin at 37 °C to OD600 of 0.6–0.8, followed by induction step with 0.5 mM IPTG at 25 °C overnight. Cells were collected and proteins were extracted by sonication with lysis buffer. The lysate was clarified by centrifugation and incubated with GST-agarose beads (company) for 2 h at 4 °C. The bound protein was washed three times and stored in PBS for the next experiments.

### 4.7. In Vitro Acetylation Assay

Acetylation assay was performed as described previously [12]. Briefly, freshly prepared recombinant ARD1/NAA10 and Hsp70 were incubated in the reaction mixture [50 mmol/L Tris-HCl (pH 8.0), 0.1 mmol/L EDTA, 1 mmol/L DTT, 10% glycerol, and 1 mmol/L acetyl-CoA at final concentration or at indicated concentration of acetyl-CoA] at 37 °C at an indicated time. The reaction was stopped by sample buffer, subsequently loaded onto SDS-PAGE for analysis by immunoblotting. To study the effect of pH on the in vitro acetylation, the reactions were performed in a range of buffer pH from 6.5 to 9.0 in the final concentration of 1 mM acetyl-CoA.

### 4.8. Immunoblotting

The SDS-PAGE gel was transferred to nitrocellulose membranes (Amersham Bioscience, Buckinghamshire, UK) for Western blot analysis. Detection was performed using ECL (enhanced electrochemiluminescence) plus (Thermo Scientific, Rockford, IL, USA).

### 4.9. DTNB-Based Nt-Acetylation Assay

1 μM purified proteins were subjected to the reaction mixture [50 mmol/L Tris-HCl (pH 8.0), 0.2 mmol/L EDTA1, 0.4 mmol/L acetyl-CoA] with the presence of 0.4 mmol/L peptides. After 30 min incubation at 37 °C, the quenching buffer (100 mM Na_2_HPO_4_ (pH 6.8), 6.4M quinidine hydrochloride) was added to stop the reaction, followed by the addition of 5,5′–dithiobis-(2-nitrobenzoic acid) (DTNB) buffer (100 mM Na_2_HPO_4_ (pH 6.8), 10 mM EDTA, 8 mM DTNB). The product formation was analyzed by spectrophotometer at 412 nm. The sequence of the peptides presenting N-termini of γ-actin and β-actin were, respectively, as follows: EEEIAALRWGRPVGRRRRPVRVYP, DDDIAALRWGRPVGRRRRPVRVYP.

## 5. Conclusions

Collectively, we proved and characterized the KAT activity of rhARD1/NAA10 in vitro, and also clarified the specific experimental conditions that are suitable for investigating KAT activity of rhARD1/NAA10 in vitro. Our findings provide a clue that possibly explains the complicated and controversial results of recent studies on KAT activity of hARD1/NAA10. Moreover, as hARD1/NAA10 stands out as an attractive drug target for cancer treatment, our finding may be helpful for the development of hARD1/NAA10 inhibitors, which can be applied to cancer therapy.

## Figures and Tables

**Figure 1 molecules-25-00588-f001:**
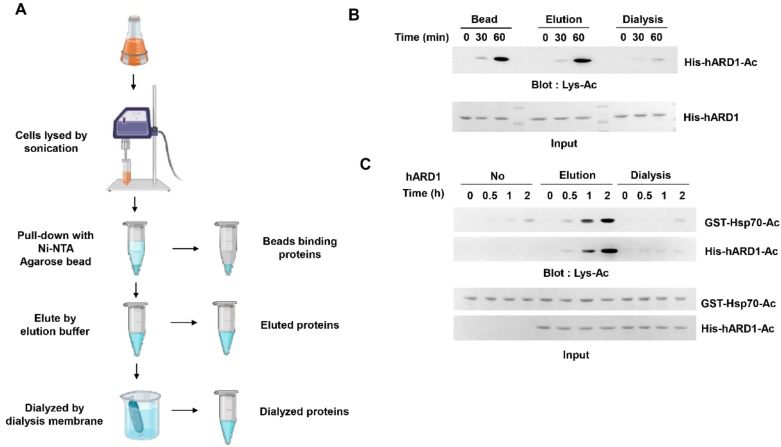
rhARD1/NAA10 protein loses its KAT activity during dialysis. (**A**) Scheme of small-scale purification of hARD1/NAA10. Ni-NTA agarose beads were incubated with lysates from harvested cells for 2 h, and the protein was eluted from beads with an elution buffer and subjected to dialysis with a dialysis membrane. (**B**) Autoacetylation activity of hARD1/NAA10 is lost during dialysis. His-tagged hARD1/NAA10 recombinants collected at different steps of purification were subjected *to* in vitro acetylation assays at indicated times, and autoacetylation level was analyzed with immunoblotting using an anti-Lys-Ac antibody. (**C**) The lysine acetyltransferase activity of hARD1/NAA10 toward Hsp70 is lost after dialysis. In vitro acetylation assays of His-hARD1/NAA10 and GST-Hsp70 at different time points were performed and assessed with western blotting using an anti-Lys-Ac antibody.

**Figure 2 molecules-25-00588-f002:**
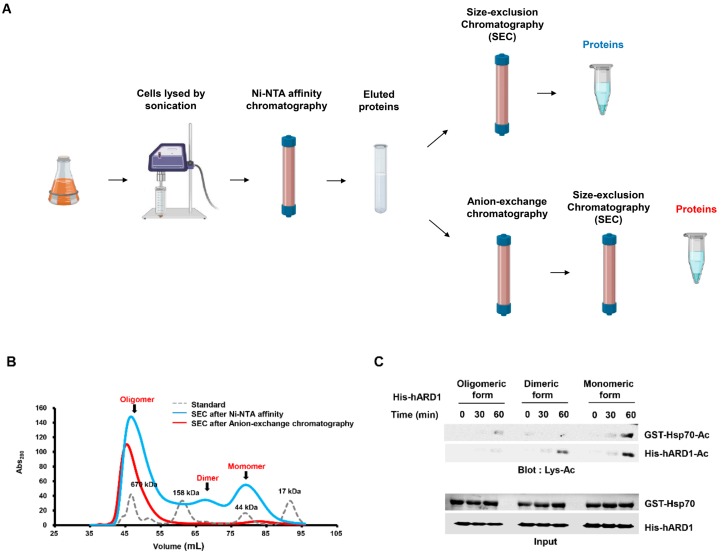
hARD1/NAA10 forms high-order aggregates during purification. (**A**) Scheme of large-scale purification of hARD1/NAA10. Lysates from harvested cells were loaded onto a Ni-NTA column, and then subjected to anion-exchange purification with HiTrap Q HP column and size-exclusion chromatography, as described in Materials and Methods. (**B**) Oligomeric status of His-hARD1/NAA10 recombinants from eluted affinity fractions and anion exchange fractions are examined with size-exclusion chromatography. (**C**) Highly oligomeric His-hARD1/NAA10 loses its KAT activity. KAT activities of various oligomeric forms of His-hARD1/NAA10 were measured using the oligomeric forms of eluted hARD1/NAA10 in the in vitro acetylation assay at indicated time points. Acetylation levels of Hsp70 and hARD1/NAA10 were assessed with western blotting. (**D**) The NAT activity of different oligomeric forms of hARD1/NAA10. (Left) Different oligomeric states of His-hARD1/NAA10 were subjected to in vitro N-terminal acetylation assay using EEEIA peptide representing N-teminus of γ-actin. The activity of monomer form is defined as 100%. (Right) Different oligomeric states of His-hARD1/NAA10 were subjected to in vitro N-terminal acetylation assay using DDDIA peptide representing N-teminus of β-actin. The activity of monomer form is defined as 100%. The experiments were performed in triplicate. The error bar indicates S.E.M. * *p* < 0.05; ** *p* < 0.01; n.s: not significant.

**Figure 3 molecules-25-00588-f003:**
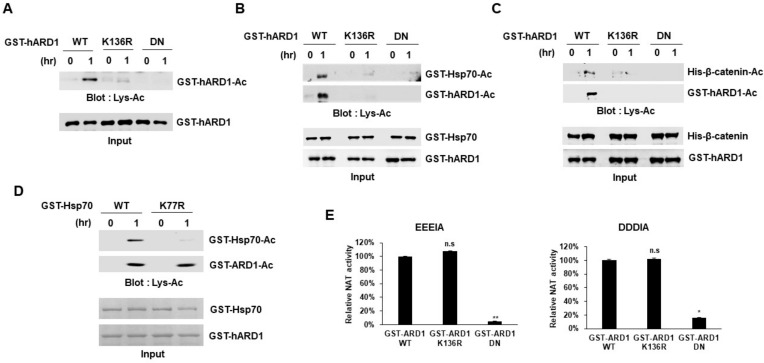
rhARD1/NAA10 protein exhibits site-specific lysine acetyltransferase activity toward its substrate. (**A**) Autoacetylation of hARD1/NAA10 specifically occurs at the residue K136. GST-hARD1/NAA10 WT and its mutants, including K136R and DN, were subjected to in vitro acetylation assay and the acetylation levels were measured with western blot analysis. WT, wild-type; DN, dominant-negative mutant. (**B**) Acetylation of Hsp70 is dependent on the catalytic capacity of hARD1/NAA10. GST-Hsp70 was incubated with or without GST-hARD1/NAA10 WT or GST-hARD1/NAA10 K136R or GST-hARD1/NAA10 DN for 1 h. WT, wild-type; DN, dominant-negative mutant. (**C**) Acetylation of β-catenin is dependent on the catalytic capacity of hARD1/NAA10. His-β-catenin was subjected to in vitro acetylation assay with or without GST-hARD1/NAA10 WT or GST-hARD1/NAA10 K136R or GST-hARD1/NAA10 DN for 1 h. WT, wild-type; DN, dominant-negative mutant. (**D**) hARD1/NAA10 acetylates the K77 residue of Hsp70. GST-Hsp70 WT and GST-Hsp70 K77R mutant were subjected to in vitro acetylation assay with GST-hARD1/NAA10, followed by immunoblotting with an anti-Lys-Ac antibody. WT, wild-type. (**E**) NAT activity of hARD1/NAA10 is not affected by autoacetylation capacity. Recombinant hARD1/NAA10 WT and mutants including K136R and DN were incubated with EEEIA and DDDIA peptide substrates in a DTNB-based Nt-acetylation assay. The NAT activity of hARD1/NAA10 WT is defined as 100%. The experiments were performed in triplicate. The error bar indicates S.E.M. * *p* < 0.05; ** *p* < 0.01; n.s, not significant.

**Figure 4 molecules-25-00588-f004:**
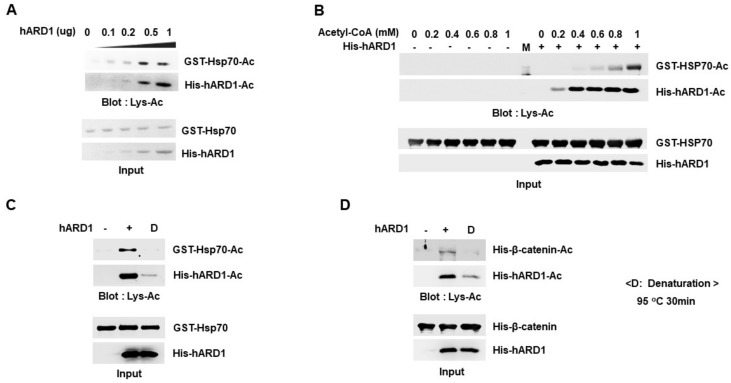
hARD1/NAA10-mediated enzymatic lysine acetylation is dependent on concentration of reactants. (**A**) Acetylation of Hsp70 occurs in a hARD1/NAA10 dose-dependent manner. Hsp70 recombinant protein was applied to in vitro acetylation assay in the presence of increasing concentrations of rhARD1/NAA10 protein for 1 h at 37 °C. (**B**) Acetylation reaction mediated by hARD1/NAA10 is influenced by the concentration of acetyl-CoA. Hsp70 recombinant protein was incubated with or without hARD1/NAA10 in the presence of increasing concentrations of acetyl-CoA for 1 h at 37 °C. M, protein marker. (**C**) KAT activity of hARD1/NAA10 is lost under harsh denaturation conditions. rhARD1/NAA10 protein was denatured at 95 °C for 30 min prior to its incubation for in vitro acetylation assay in the presence of Hsp70 recombinant protein for 2 h at 37 °C. Western blot analysis was carried out to evaluate the acetylation status of proteins. (**D**) rhARD1/NAA10 was heated at 95 °C for 30 min prior to being subjected to in vitro acetylation assay in the presence of β-catenin. Acetylation levels were measured by western blot analysis.

**Figure 5 molecules-25-00588-f005:**
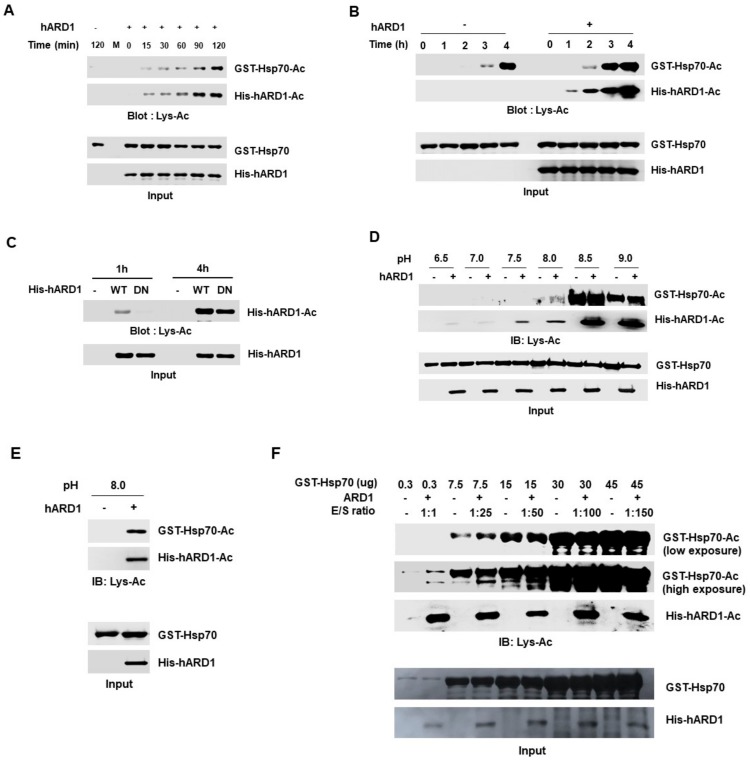
hARD1/NAA10-mediated lysine acetylation occurs under limited reaction conditions. (**A**) Autoacetylation activity of hARD1/NAA10 is dependent on the reaction time. A time-dependent in vitro acetylation assay was performed and western blot analysis was carried out to analyze the levels of acetylated protein. (**B**) hARD1/NAA10-mediated lysine acetylation in vitro is limited to a reaction time of 2 h. Hsp70 recombinant protein was incubated for different time points without or with purified rhARD1 protein. The acetylated levels were measured with immunoblotting. (**C**) Non-enzymatic acetylation reaction occurs at long incubation time. Autoacetylation of hARD1/NAA10 was analyzed with western blotting after 1 and 4 h incubation of hARD1/NAA10 WT and hARD1 DN mutant in in vitro acetylation assay. WT, wild-type; DN, dominant-negative mutant. (**D**) hARD1/NAA10-mediated lysine acetylation in vitro is favorable at pH 8.0. Hsp70 was subjected to in vitro acetylation assay in various pH with or without hARD1/NAA10. (**E**) hARD1/NAA10-mediated lysine acetylation of Hsp70 at pH 8.0. (**F**) Limited ratio of hARD1/NAA10 and Hsp70 to distinguish the enzymatic activity and the non-specific chemical reaction. The in vitro acetylation assays were performed in the varied ratio between hARD1/NAA10 and Hsp70.

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
