# Peer review of "Characterization of Lysine Acetyltransferase Activity of Recombinant Human ARD1/NAA10"

_molecules, 2020, doi:10.3390/molecules25030588_

Round 1
Reviewer 1 Report
In their revision, Tam and coworkers addressed most concerns, resulting in an improved manuscript. However, few issues from the new manuscript still remain:
1. The comment #2 has not been addressed. The NAA10/NAA15 co-expressed complex with correct stoichiometry purified from E. coli has been reported by other groups. This control sample is essential to avoid potential experimental artifacts because the complex is a physiological state of NAA10.
2. In the page 7 line 258, the authors stated that limited reaction conditions were identified for distinguish the hARD1/NAA10-lysine acetylation from non-specific chemical reaction. However, only one factor—reaction time was checked. This needs more extensive experiments, for example pH, buffer and E/S ratio.
3. Page 6, line 229, there is no citation for the ‘previous report’.
Author Response
"Please see the attachment."

Reviewer 2 Report
All my concerns have been addressed.
Author Response
We would like to express our appreciation to reviewer for spending precious time reviewing our manuscript. The valuable comments from the reviewer definitely help us improve our manuscript.
This manuscript is a resubmission of an earlier submission. The following is a list of the peer review reports and author responses from that submission.
Round 1
Reviewer 1 Report
‘Characterization of lysine acetyltransferase activity of recombinant human ARD1’ by Vo et al. presents data on the in vitro lysine acetyltransferase activity of ARD1/NAA10.
The manuscript is a timely contribution to an ongoing debate on the activity and specificity of the acetyltransferase ARD1/NAA10. ARD1/NAA10 is an evolutionarily conserved N-terminal acetyltransferase (NAT) and the central question is whether and under which conditions ARD1/NAA10 is able to acetylate lysine residues of various proteins (thus acting as a lysine acetyltransferase, KAT). Several lysine substrates of ARD1/NAA10 have been reported, some with considerable impact on embryonic development, cancer development and physiology. However, some of these lysine acetylation events have not been reproduced by independent efforts, and structural data indicates that ARD1/NAA10 is mainly a NAT enzyme.
In the current manuscript, Vo and colleagues have attempted a systematic test of the conditions under which ARD1/NAA10 acetylates Hsp70, which is acetylated by ARD1/NAA10 on K77. They have tested the effect of different purification schemes on recombinant ARD1/NAA10 activity towards recombinant Hsp70, as well as the effect of enzyme dose and temperature.
Suggestions for improvements:
1) ARD1 is the old gene/protein name while the new official gene/protein name approved by HUGO Gene Nomenclature Committee (HGNC) is NAA10 (N(alpha)-acetyltransferase 10). Thus NAA10 should be used throughout the paper, and both names should be listed initially.
2) One concern is the assay conditions used. Magin et al. (reference #15 in the manuscript) showed that the acetyl CoA concentration in reaction setups drive nonenzymatic acetylation. While Magin et al., have used up to 1 mM acetyl CoA to illustrate their point about nonenzymatic acetylation, the present assays use 10 mM. This is in the order of 10-100 times the concentration even in mitochondria, and potentially 1000 times the cytosolic concentration, which is the environment relevant for ARD1/NAA10. Can the authors show that the Hsp70 acetylation takes place closer to the actual cellular concentrations? Using a peptide based assay (DTNB or 14C-Acetyl-Coenzyme A assays for example) for this would be helpful (I realize that getting a good signal in their Western blot based assay may be challenging with low substrate doses). This could also shed light on the specificity of the reaction and the possibility of other Lys sites being an acetyl acceptor (the top right band in Fig. 3C suggests this may be the case unless this is an uncatalyzed event).
3) In Figure 1 it is shown that dialyzed ARD1/NAA10 loses most of its KAT activity towards Hsp70. The implication seems to be that their dialysis explains the loss of KAT activity in Magin et al.. However, the enzyme purified by Magin et al. is very active as a NAT (even at substrate concentrations less than a tenth of the ones used here, and at much shorter reaction time). So I wonder whether Vo et al. have looked at NAT activity? It is important to assess whether or not ARD1/NAA10 is overall catalytically dead or just KAT-dead under the conditions tested. Thus, for selected conditions, NAT-activity should be tested in parallel to Hsp70 Lys-acetylation. Regarding the loss of activity of protein aggregates/oligomers (Fig. 2B), this is commonly observed for purified enzymes and this is not the fraction that is used for activity assays in most labs.
4) In Fig. 3C, initial acetylation of Hsp70 K77R seems to be equal to Hsp70 WT at 30 mins. Also, the level of ARD1/NAA10 acetylation seems to be lower for K77R than Hsp70 WT (at 1h). This should be repeated and explained.
5) In Fig. 3 it would be interesting to include the ARD1/NAA10 K136R mutant towards Hsp70 as an additional control.
6) Figure 4C. The heat inactivated ARD1/NAA10 still mediates some Hsp70 acetylation. Is the enzyme completely denatured? Are there other components in this reaction that may explain this?
7) In Figure 5B – Hsp70 acetylation levels are varying in a somewhat strange manner. For instance, at 3h Hsp70 acetylation is weaker when ARD1/NAA10 is added (as compared to when it is not added) while at 4h addition of ARD1/NAA10 gives more Hsp70 acetylation. Perhaps this is due to differences in Hsp70 addition/loading? This should be corrected.
8) Recently, the KAT activity of ARD1/NAA10 was proposed to be induced by cellular hydroxylation of ARD1/NAA10 by FIH1 (Kang et al., Redox Biol, 2018, PMID: 30237125). Could the authors please discuss their data in light of these findings?
Reviewer 2 Report
Tam Thuy Lu Vo et al. provided a few pieces of evidence to show that recombinant expressed human Arrest defective 1 (ARD1) exhibited the lysine acetyltransferase (KAT) activity in vitro, which is currently considered as an artificial chemical reaction caused by reactants. They reasoned that the oligomerization state of ARD1 is closely correlated with its KAT activity, which is shown in ARD1 monomer but not in oligomer state. The authors also defined some experimental conditions for ARD1 to retain its KAT activity. Overall, this study should be potentially helping to revise the current views on the ARD1 protein. However, stronger evidence, including several additional experiments/controls should be required to justify some conclusions.
Major issues:
The recombinantly expressed ARD1 in E. coli easily loses its KAT activity or KAT activity is very weak in vitro. It is generally acceptable that some mammalian proteins expressed in E. coli may have no enzymatic activity. Therefore, whether the intrinsic instability of hARD1 is caused by the expression systems should be addressed by comparing its prep purified from mammalian cells.It is well known that ARD1 is not stable when expressed by its own. A good control experiment is to purify this subunit by co-expressing with its auxiliary subunit Naa15p, which is possibly removed after purification.
Fig.1 concludes that the eluted hARD1 protein maintains the KAT activity, while the activity is lost after dialysis. However, only KAT activity of hARD1 was fully examined. A control experiment like the NAT activity of hARD1 also needs to be checked. Similarly, the examination of the NAT activity of hARD1 also should be included in the Figs. 2 and 4.
Fig.1 uses a purified protein only from Ni-NTA affinity column. This protein prep is generally not considered pure enough for many experiments. How can authors sure these observations were not caused by any contaminated bacterial proteins? In light of data in Fig. 2, where the KAT activity is only barely detectable when using all gel filtration fractions (aggregated, dimeric and monomeric), a negative control, for example, hARD1 mutants, should be included.
The author rationalized that the aggregation or high oligomerization of hARD1 caused the loss of KAT activity in Fig. 2. Both of His tagged and GST tagged hARD1 were stored in a PBS buffer indicated in method section. From the profile of size-exclusion chromatography, either hARD1 from Ni-NTA affinity or after Anion-exchange chromatography showed that the large amount of hARD1 was in aggregation state, suggesting the protein was not in corrected conformation. This may be caused by the PBS buffer that is not suitable for ARD1, or just by its intrinsically unstable feature. In addition, His-tagged hARD1 was used in this experiment, and how about GST-tagged hARD1 as compared to this?
GST-hARD1 was used to perform lysine acetyltransferase activity toward Hsp70 in Fig.3. For the similar lysine acetyltransferase assay shown in Fig.1, 2, 4, 5, however, His-hARD1 was used. Is this a labelling mistake? If not, the authors should give their reasons.
It is extremely odd to see significant amount of activity of this unstable hARD1 on Hsp70 after temperature treatments in Fig. 4. There was even some autoacetylation activity after 30 min at 65C, even though it does not make difference in the Hsp70 acetylation. This experiment should be redesigned to make a conclusion on temperature dependence.
Both GST-Hsp70 and His-hARD1 showed degradation with increased time and only a small amount of protein was left after 4h shown in Fig. 5B. Oddly, however, the signal of acetylation of Hsp70 and hARD1 was increased and even reached a highest level after 4h. The reason for this weird results should be emphasized!!!! This degradation of GST-Hsp70 was only observed in the presence with His-hARD1 not for itself. What is a reason for this degradation? Moreover, His-hARD1 did not degrade after 4h shown in Fig. 5C, which is not consistent with the data in Fig. 5B.
All lysine acetyltransferase activity assays in the manuscript just used only one protein Hsp70 as a substrate of hARD1. To strengthen the conclusions, one more substrate should be used to repeat this assays, like HIF-1α, β-catenin and PGK1 mentioned in the introduction section.
Minor issues:
Line 94: There is no citation for this “previous report”. Line 381: What is the component of dialysis buffer for the eluted protein? Line 185: “As shown Figure 4a” should be “As shown in Figure 4a”. Fig. 4B: What is a mysterious band in the middle of gel shown GST-HSP70-Ac? From the input gel, there are no protein samples in this position. The authors should explain this. Mixed labels: Ac-Lys vs Lys-Ac vs Lysz-Ac. Should be consistent. CoA used in Fig. 4b?